# Quantification of the Influence of Citrate/Fe(II) Molar Ratio on Hydroxyl Radical Production and Pollutant Degradation during Fe(II)-Catalyzed O_2_ and H_2_O_2_ Oxidation Processes

**DOI:** 10.3390/ijerph191912977

**Published:** 2022-10-10

**Authors:** Bingbing Hu, Peng Zhang, Hui Liu, Songhu Yuan

**Affiliations:** 1State Key Laboratory of Biogeology and Environmental Geology, China University of Geosciences, 68 Jincheng Street, East Lake High-Tech Development Zone, Wuhan 430078, China; 2Hubei Key Laboratory of Yangtze Catchment Environmental Aquatic Science, School of Environmental Studies, China University of Geosciences, 68 Jincheng Street, East Lake High-Tech Development Zone, Wuhan 430078, China

**Keywords:** hydroxyl radicals, Fe(II), citrate, molar ratio, oxidation

## Abstract

Ligand-enhanced hydroxyl radical (•OH) production is an important strategy for Fe(II)-catalyzed O_2_ and H_2_O_2_ oxidation processes. However, the influence of the molar ratio of ligands to Fe(II) on •OH production remains elusive. This study employed citrate and inorganic dissolved Fe(II) (Fe(II)_dis_) as the representative ligand and Fe(II) species, respectively, to quantify this relationship. Results showed that •OH production was highly dependent on the citrate/Fe(II) molar ratio. For instance, for the oxygenation of Fe(II)_dis_, the •OH accumulations were 2.0–8.5, 3.4–28.5 and 8.1–42.3 μM at low (0.25–0.5), moderate (0.5–1), and high (1–2) citrate/Fe(II) molar ratios, respectively. At low citrate/Fe(II) molar ratio (<0.5), inorganic Fe(II)_dis_ mainly contributed to •OH production, with the increase in the citrate/Fe(II) molar ratio to a high level (1–2), Fe(II)-citrate complex turned to the electron source for •OH production. The change in Fe(II) speciation with the increase of citrate/Fe(II) molar ratio elevated •OH production. For pollutant degradation, 1 mg/L phenol was degraded by 53.6% within 40 min during oxygenation of Fe(II)-citrate system (1:1) at pH 7. Our results suggest that a moderate molar ratio of ligand/Fe(II) (0.5–1) may be optimal for Fe(II)-catalyzed O_2_ and H_2_O_2_ oxidation processes.

## 1. Introduction

Hydroxyl radical-based advanced oxidation process (HR-AOP) is considered to be an effective technology for the remediation of contaminated soil and groundwater [1]. The hydroxyl radical (•OH) is the most powerful oxidant in the natural system [2,3,4], capable of oxidizing most environmental pollutants, such as chlorinated hydrocarbons [5,6], antibiotic [7] and aromatic compounds [8,9,10]. Hydrogen peroxide (H_2_O_2_) is the traditional source of •OH in HR-AOP, and recent studies showed that oxygen (O_2_) can also act as a source of •OH in some specific conditions, for example, when the reduced soil or sediment was exposed to oxic conditions [1,6,7]. However, H_2_O_2_ and O_2_ cannot be spontaneously transformed to •OH in natural environments, requiring activation by chemical agents or physical treatments. Ferrous iron (Fe(II)) is the most commonly used and effective activator for H_2_O_2_ and O_2_ [1,11,12]. The Fe(II)-catalyzed H_2_O_2_ oxidation process (i.e., Fenton reaction) has been widely used in contaminant remediation, such as in situ chemical oxidation projects and wastewater treatment [1,11]. In comparison, the Fe(II)-catalyzed O_2_ oxidation process has been demonstrated to degrade pollutants only at the laboratory scale [5,6,7,8,9,13]. However, O_2_ is cheaper and easier to obtain than H_2_O_2_, so the Fe(II)-catalyzed O_2_ oxidation process is also proposed as a promising method for pollutant degradation [1].

Although the Fe(II)-catalyzed H_2_O_2_ oxidation process is effective in acidic pH conditions, the efficiency of pollutant degradation was poor at circumneutral or alkaline pH conditions [6,7,8,9,11,13,14,15]. This change was related to the variation of Fe(II) and Fe(III) speciation at a different solution pH. With the increase of solution pH, Fe(II) speciation varied from Fe^2+^ to FeOH^+^ and Fe(OH)_2_, resulting in the decrease of •OH yield from H_2_O_2_ decomposition by Fe(II) at high pH conditions [12]. In addition, inorganic Fe^3+^ will hydrolyze to Fe(III) precipitation at high pH conditions, which is not conducive to Fe(II) regeneration [11]. For the Fe(II)-catalyzed O_2_ oxidation process, the •OH yield and efficiency of pollutant degradation were relatively low at circumneutral pH conditions [6,7,9,13].

To enhance the •OH yield during Fe(II)-catalyzed O_2_ and H_2_O_2_ oxidation processes, a common strategy is adding ligands (L, such as citrate) to regulate Fe(II)/Fe(III) speciation [11]. A large number of studies have shown that the addition of ligands can increase •OH yield during Fe(II)-catalyzed O_2_ and H_2_O_2_ oxidation processes and improve pollutant degradation efficiency [11,16,17,18,19]. For instance, Xie et al. (2021) reported that the addition of 2 mM tripolyphosphate elevated trichloroethylene degradation from 13% to 80% in oxic sediment suspension [16]. Given the significance of ligand-enhanced •OH production during Fe(II)-catalyzed O_2_ and H_2_O_2_ oxidation processes, many studies have explored the mechanisms of ligand-enhanced •OH production and found that the enhancement was related to the formation of Fe(II)/Fe(III)-L complexes through the complexation of Fe(II)/Fe(III) by ligands [11,12,16,17,18,19,20,21,22,23]. In general, the higher concentrations of Fe(II)/Fe(III)-L complexes result in a more significant enhancement in •OH production during Fe(II)-catalyzed O_2_ and H_2_O_2_ oxidation processes [1,11,23].

Since the generation of Fe(II)-L complex depended on the concentrations of ligand and free dissolved Fe(II) (Fe(II)_dis_) and the complexation ability of ligands [18], the influence of ligand concentrations and types and Fe(II) dosages on •OH production and pollutant degradation were widely investigated [16,17,18,20,21,24]. However, there was no conclusive agreement on the roles of the molar ratio of ligand/Fe(II) on •OH production and pollutant degradation [11]. Because the essence of changing ligand and Fe(II) dosages in different reaction systems is to regulate the molar ratio of ligand/Fe(II), the latter is a key factor for Fe(II)-catalyzed O_2_ and H_2_O_2_ oxidation processes. These knowledge gaps limit the development of Fe(II)-catalyzed O_2_ and H_2_O_2_ oxidation processes.

The objective of this study was to quantify the influence of ligand/Fe(II) ratio on •OH production and pollutant degradation during Fe(II)-catalyzed O_2_ and H_2_O_2_ oxidation processes. To quantify •OH production during the reaction process, the oxidation of benzoate to *p*-hydroxybenzoic acid (*p*-HBA) was used as a probe reaction [25,26]. For the sake of simplicity, citrate and inorganic Fe(II)_dis_ were chosen as the representatives of ligands and Fe(II) species, respectively. Note that both citrate and inorganic Fe(II)_dis_ have been widely used in previous studies [16,18,21,23,24]. The influence of citrate/Fe(II) molar ratio on •OH production was assessed by fixing the initial Fe(II) concentration while varying the initial citrate concentration over a pH range 6–7.5. Phenol was chosen as a model pollutant to test the oxidative impact of an Fe(II)-O_2_ system. A kinetic model was developed to quantitatively describe the dependence of •OH production and pollutant degradation on citrate/Fe(II) molar ratio.

## 2. Materials and Methods

### 2.1. Chemicals

Ferrous sulfate heptahydrate (FeSO_4_·7H_2_O, 99.9%), trisodium citrate dihydrate (C_6_H_5_Na_3_O_7_·2H_2_O, 99%), sodium benzoate (99.5%), *p*-hydroxybenzoic acid (*p*-HBA, 99%) and boric acid were purchased from Sinopharm Chemical Regent Co. Ltd., China. Piperazine-N, N-bis-(ethanesulfonic acid) sodium salt (PIPES, 99%) and 2-(N-morpholino)ethanesulfonic acid (MES, 99%) were acquired from Sigma-Aldrich. All other chemicals were of analytical grade or above. Deionized water (18.2 MΩ·cm) produced by a Heal Force NW ultrapure water System was used in all the experiments.

### 2.2. Oxic Experiments

A series of oxic experiments were used to explore the influence of citrate/Fe(II) molar ratio on •OH production and phenol degradation during the Fe(II)-catalyzed O_2_ oxidation process. All oxic experiments were conducted at 25 ± 1 °C in 150-mL conical flasks that were wrapped with aluminum foil to avoid light. Teflon-coated magnetic stirring bars were used to keep the stirring speed at approximately 300 rpm. To initiate experiments, citrate concentrations of 62.5, 125, 250, and 500 μM were added to the solution containing 250 μM Fe(II)_dis_, 10 mM buffer (MES for pH 6 and PIPES for pH 7 and 7.5), 20 mM benzoate and 5 mM Na_2_SO_4_. Control experiments were carried out with the addition of inorganic Fe(II)_dis_ alone under otherwise identical conditions.

To evaluate the oxidative impact of •OH produced from oxygenation of Fe(II)-citrate system, 1 mg/L phenol (*k*_phenol, •OH_ = 6.6 × 10^9^ M^−1^ s^−1^, [27]) was added to the Fe(II)-citrate solution containing 250 μM citrate, 250 μM Fe(II)_dis_ and 100 mM boric acid. It is noted that boric acid had a marginal influence on pollutant degradation [5] due to the low reaction rate constant of boric acid with •OH (*k*_boric acid, •OH_ < 5 × 10^4^ M^−1^ s^−1^, [27]).

During all of the above experiments, the change of solution pH was less than 0.1 and the dissolved O_2_ (DO) concentration was maintained at near 0.25 mM. At predetermined times of reaction, approximately 1-mL sample was taken out for *p*-HBA, Fe(II)_dis_, Fe(III)_dis_ and total Fe analysis. All experiments were performed in duplicate.

### 2.3. Anoxic Experiments

The anoxic experiments were used to explore the influence of citrate/Fe(II) molar ratio on •OH production during the Fe(II)-catalyzed H_2_O_2_ oxidation process. A solution containing 250 μM inorganic Fe(II)_dis_, 20 mM benzoate and 10 mM buffer was first purged with N_2_ (99.999%) for at least 1 h and then mixed with different concentrations of citrate (0, 62.5, 125, 250 and 500 μM) for 2 h in an anaerobic glovebox (95% N_2_ and 5% H_2_, COY, USA). Finally, H_2_O_2_ concentrations of 20, 40, 60, 80 and 100 μM were added to the above Fe(II)-citrate system. After a reaction of 30 min, a sample of approximately 1 mL was removed for *p*-HBA. Note that the residual concentration of H_2_O_2_ in all experiments was less than 0.1 μM. These experiments were conducted at pH 6, 7 and 7.5.

### 2.4. Analysis

For the analysis of the *p*-HBA concentration, a sample of approximately 0.8 mL was rapidly mixed with 0.8 mL of methanol (HPLC grade) to quench further oxidation of benzoate by •OH and then the suspension was filtered through a membrane of 0.22 μm. The concentration of *p*-HBA was determined by HPLC according to the previous method [28]. The conversion coefficient between *p*-HBA and cumulative •OH was 5.87 [25,26]. The detection limit of *p*-HBA was 0.1 μM. For the analysis of Fe(II)_dis_ and Fe(III)_dis_, another sample of 0.8 mL was filtered through a membrane of 0.22 μm and the filtrate was collected in a pre-acidified vial. Fe(II)_dis_ concentration was measured by the ferrozine method at wavelength 562 nm [29]. To minimize Fe(III)-citrate interference, Fe(II) analysis was performed within 15 min after the chromogenic reaction. Dissolved total iron (Fe_total_) was determined by reducing Fe(III) to Fe(II) with hydroxylamine-HCl. The Fe(III)_dis_ was calculated using the difference between dissolved Fe_total_ and Fe(II)_dis_ concentrations. For the analysis of solid Fe(II) and Fe(III), the sample was directly mixed with 1 M HCl to dissolve solid phase components and then analyzed. For the analysis of Fe(III) in colloids (1–220 nm) and true solution (<1 nm), the sample was fractioned by filters of 0.22 μm and ultrafiltration membranes of 20 nm (30 kDa, Millipore) and 1 nm (3 kDa, Pall), respectively. The concentration of phenol was measured by HPLC [10].

### 2.5. Kinetic Modeling and Speciation Calculation

A kinetic model was developed to fit the concentration time series data of Fe(II)_dis_, Fe(III)_dis_ and •OH under different experimental conditions. Kintecus 6.51 software (James C. Ianni, Lansdowne, PA, USA) was used for calculation [30]. The reaction networks are shown in Table 1 and consist of two subsections: (1) inorganic Fe(II)_dis_ oxidation and (2) extended reactions in the presence of citrate. In the inorganic Fe(II)_dis_ system, both the oxidation of inorganic Fe(II)_dis_ and adsorbed Fe(II) (Fe(II)_ad_) were considered (reactions A1–A13). In Fe(II)-citrate system, the complexation of Fe(II)_dis_ and Fe(III)_dis_ by citrate (reactions B1–B4) and the oxidation of citrate complexed Fe(II) (reactions B5–B8) were considered. For the sake of simplification, the interactions among •O_2_^−^, H_2_O_2_ and •OH were not included in the kinetic model. More details are given in Appendix A. Because the solution pH varied less than 0.1 unit during experiments, constant pH was used in the calculations. DO concentration was set at 0.25 mM.

To evaluate the influence of citrate/Fe(II) molar ratio on the fraction of complexed Fe(II)/Fe(III), a speciation calculation for Fe(II) and Fe(III) under different experimental conditions was performed with Visual MINTEQ 3.1 [39].

## 3. Results and Discussion

### 3.1. Effect of Citrate/Fe(II) Molar Ratio on •OH Production during Fe(II)-Catalyzed O_2_ Oxidation Process

For oxygenation of 250 μM inorganic Fe(II)_dis_ in the absence of citrate, the concentrations of cumulative •OH were always below the detection limit at pH 6 and gradually increased to 1.3 and 0.7 μM at pH 7 and 7.5, respectively (Figure 1a–c). When citrate was added to the above inorganic Fe(II)_dis_ system, •OH accumulation rapidly elevated (Figure 1a–c). For instance, for oxygenation of 250 μM inorganic Fe(II)_dis_ in the presence of 250 μM citrate, •OH accumulation reached 28.5 (>47.5-fold), 13.3 (~10.2-fold) and 8.1 (~11.6-fold) μM for pH 6, 7 and 7.5, respectively (Figure 1), which were much higher than those in the inorganic Fe(II)_dis_ system. The enhancement of citrate on •OH production was in agreement with previous observations [16,18,19,21], which further supported the conclusion that the addition of ligands can effectively reinforce •OH production from Fe(II) oxidation by O_2_.

As shown in Figure 1d, the enhancement of citrate on •OH production was highly dependent on citrate/Fe(II) molar ratio and solution pH. At pH 6, the cumulative •OH increased from 2.8 to 8.5 μM (~3.0-fold) with increasing citrate/Fe(II) molar ratio from 0.25 to 0.5 (Figure 1d) and increased from 8.5 to 28.5 μM (~3.3-fold) with increasing citrate/Fe(II) molar ratio from 0.5 to 1 (Figure 1d). When the citrate/Fe(II) molar ratio further increased to 2, the cumulative •OH increased to 42.3 μM (~1.5-fold) (Figure 1d). At pH 7, the cumulative •OH increased from 2.0 to 4.9 μM (~2.5-fold) with increasing citrate/Fe(II) molar ratio from 0.25 to 0.5 (Figure 1d) and increased from 4.9 to 13.3 μM (~2.7-fold) with increasing citrate/Fe(II) molar ratio from 0.5 to 1 (Figure 1d). When the citrate/Fe(II) molar ratio further increased to 2, the cumulative •OH increased to 24.2 μM (~1.8-fold) (Figure 1d). At pH 7.5, the cumulative •OH increased from 2.2 to 3.4 μM (~1.5-fold) with increasing citrate/Fe(II) molar ratio from 0.25 to 0.5 (Figure 1d) and increased from 3.4 to 8.1 μM (~2.4-fold) with increasing citrate/Fe(II) molar ratio from 0.5 to 1 (Figure 1d). When the citrate/Fe(II) molar ratio further increased to 2, the cumulative •OH increased to 18.0 μM (~2.2-fold) (Figure 1d). The increased folds on •OH accumulation suggests that a moderate citrate/Fe(II) molar ratio can result in a more significant increase in •OH production.

### 3.2. Effect of Citrate/Fe(II) Molar Ratio on •OH Yield during Fe(II)-Catalyzed H_2_O_2_ Oxidation Process

Similar to the Fe(II)-catalyzed O_2_ oxidation process, the presence of citrate also facilitated •OH production during the Fe(II)-catalyzed H_2_O_2_ oxidation process (Figure 2a–c). For instance, for the oxidation of 250 μM Fe(II) by 100 μM H_2_O_2_ at pH 7, the •OH accumulation increased from 1.5 to 28.7 μM (~19.1-fold) when the citrate concentration increased from 0 to 500 μM (Figure 2b). Because •OH accumulation increased linearly with increasing H_2_O_2_ concentration at a given solution pH and an initial Fe(II)_dis_ dosage, the yield of •OH relative to H_2_O_2_ decomposition can be derived from the slope of the linear fitting. Figure 2d shows that the •OH yield depended on citrate/Fe(II) molar ratio and solution pH and this dependence can also be divided into three subsections. At pH 6, the •OH yield increased from 6.3% to 7.6% (~1.2-fold) with increasing citrate/Fe(II) molar ratio from 0.25 to 0.5 and increased from 7.6% to 26.0% (~3.4-fold) with increasing citrate/Fe(II) molar ratio from 0.5 to 1 (Table 2). When the citrate/Fe(II) molar ratio further increased to 2, the •OH yield increased to 52.2% (~2-fold) (Table 2). At pH 7, the •OH yield increased from 2.9% to 4.5% (~1.6-fold) with increasing citrate/Fe(II) molar ratio from 0.25 to 0.5 and increased from 4.5% to 12.8% (~2.8-fold) with increasing citrate/Fe(II) molar ratio from 0.5 to 1 (Table 2). When the citrate/Fe(II) molar ratio further increased to 2, the •OH yield increased to 31.5% (~2.5-fold) (Table 2). At pH 7.5, the •OH yield increased from 2.9% to 3.7% (~1.3-fold) with increasing citrate/Fe(II) molar ratio from 0.25 to 0.5 and increased from 3.7% to 9.1% (~2.5-fold) with increasing citrate/Fe(II) molar ratio from 0.5 to 1 (Table 2). When citrate/Fe(II) molar ratio further increased to 2, the •OH yield increased to 17.6% (~1.9-fold) (Table 2). These results indicate that a moderate citrate/Fe(II) molar ratio was more beneficial to the increase in •OH yield from H_2_O_2_ decomposition.

Previous studies have reported that •OH yield from H_2_O_2_ decomposition by inorganic Fe(II)_dis_ at pH 7 was 1.5–1.6% [18,40], close to our measured value of 1.8% (Table 2). In the presence of citrate, a previous study reported that the •OH yield from H_2_O_2_ decomposition by Fe(II)_dis_ at pH 7 was 10% when the molar ratio of citrate/Fe(II) was 1 [18], which was also roughly consistent with the result of this study (12.8%, Table 2). At fixed Fe(II) and citrate concentrations, •OH yield decreased with increasing the solution pH from 6 to 7.5 (Table 2), which was in line with the previous observation that acidic pH conditions favored •OH production but alkaline pH was unfavorable [12]. Therefore, under similar conditions, our measurements are comparable with previous studies.

### 3.3. Variation of Fe(II)/Fe(III) Species at Different Citrate/Fe(II) Ratios during Oxidation Process

Since Fe(II) is the main electron source for •OH production in Fe(II)-citrate systems [18], the variations of Fe(II)_dis_ and Fe(III)_dis_ during oxygenation were measured. In the absence of citrate, the inorganic Fe(II)_dis_ concentrations within 20–60 min varied by <5%, 98% and 99% at pH 6, 7 and 7.5, respectively (Appendix A). Inorganic Fe(III)_dis_ was not measured, but abundant Fe(III) precipitates (>0.22 μm) were generated (Appendix A). XRD analysis shows that lepidocrocite was the main mineral phase of Fe(III) precipitates (Appendix A). However, the addition of citrate remarkably changed the processes of Fe(II)_dis_ oxidation and Fe(III)_dis_ precipitate (Figure 3). At pH 6, the presence of citrate significantly accelerated Fe(II) oxidation and the oxidation rate increased with increasing citrate/Fe(II) molar ratio (Figure 3a). At pH 7, the presence of citrate accelerated Fe(II)_dis_ oxidation only at the initial stage, i.e., there was no lag of Fe(II) oxidation within the initial 10 min, while it inhibited Fe(II)_dis_ oxidation at the last stage (Figure 3b). At pH 7.5, the addition of citrate inhibited Fe(II)_dis_ oxidation (Figure 3c). At pH 7 and 7.5, the inhibition of citrate on Fe(II) oxidation decreased with an increasing citrate/Fe(II) molar ratio. The influence of citrate on Fe(II) oxidation was related to the formation and oxidation of complexed Fe(II)/Fe(III) species (for details, see Appendix A). In addition to Fe(II)_dis_ oxidation, the presence of citrate also had a great influence on the hydrolysis and precipitate of Fe(III)_dis_. At pH 6–7.5, no Fe(III) precipitate (<1 μM) was detected in Fe(II)_dis_-citrate system (Figure 3d–f), which was opposite to the observation in the inorganic Fe(II)_dis_ system (Appendix A). Results of Appendix A show that Fe(III)_dis_ was mainly (>90%) in colloidal form (1–220 nm), while the fraction of true Fe(III)_dis_ (<1 nm) was less than 10%. Because Fe(II) oxidation by H_2_O_2_ is rapid and not easily sampled, the kinetics of Fe(II) oxidation and Fe(III)_dis_ precipitate during the Fe(II)-catalyzed H_2_O_2_ oxidation process were not experimentally measured in this study.

### 3.4. Controlling Mechanisms of Citrate/Fe(II) Molar Ratio on •OH Production

Based on the above mentioned, a kinetic model was developed to describe •OH production and Fe(II)/Fe(III) transformation during Fe(II)-catalyzed O_2_ and H_2_O_2_ oxidation processes (Table 1). As shown in Figure 1, Figure 3 and Appendix A, the model-predicted time trajectories of Fe(II)_dis_, Fe(III)_dis_ and •OH were in general agreement with the observed trends. Hence, the reactions in Table 1 can be used to describe the most important reactions in inorganic Fe(II)_dis_ and Fe(II)-citrate systems. Besides, the assumptions made in this study are reasonable and their influence on modeling results could be ignored.

To assess the relative importance of each reaction on •OH production during Fe(II)-catalyzed O_2_ and H_2_O_2_ oxidation processes, the matrices of normalized sensitivity coefficients (NSCs) at different reaction times in inorganic Fe(II)_dis_ and Fe(II)-citrate systems were calculated (Figure 4). The positive NSC values mean that reactions produce •OH, whereas the negative NSC values mean that reactions consume •OH. For the oxygenation of Fe(II)_dis_ by O_2_, in the absence of citrate, reactions A1 and A10 yielded the largest positive NSC values (Figure 4a), which confirmed the contribution of the oxidation of inorganic Fe(II)_dis_ and Fe(II)_ad_ to •OH production. However, in the presence of citrate, the decomposition of H_2_O_2_ by citrate complexed Fe(II) (reaction C7) changed, becoming the most important reaction on •OH production given the largest positive NSC value (Figure 4b). In comparison, the reaction of H_2_O_2_ with an inorganic Fe(II)_dis_ (reaction A3) generated the negative NSC value (Figure 4b). In other words, reaction A3 was a consumer of •OH. This may be unexpected given that reaction A3 can generate •OH. The explanation is that when H_2_O_2_ is decomposed by inorganic Fe(II)_dis_, less H_2_O_2_ can react with citrate complexed Fe(II), while the latter can produce more •OH (Table 1). In addition, the oxidation of Fe(II)-citrate complex (reaction C5) yielded a negative NSC value (Figure 4b), which may be explained by the fact that when the Fe(II)-citrate complex was oxidized by O_2_, less Fe(II)-citrate complex can react with H_2_O_2_ to generate •OH. Hence, Fe(II)-citrate complex is the main electron contributor for •OH production in the presence of citrate. For the oxidation of Fe(II)_dis_ by H_2_O_2_, the reactions C1 and C7 also generated the largest positive NSC values (Figure 4d), indicating that the oxidation of Fe(II)-citrate complex by H_2_O_2_ mainly contributed to •OH production in the presence of citrate.

To obtain further insight into the influence of citrate/Fe(II) molar ratio on Fe(II) species, the speciation calculation and kinetic models were executed at pH 6–7.5. Before oxidation, the fractions of Fe(II)-citrate^−^ in total Fe(II) were 24.2–49.6%, 48.3–94.2% and 88.8–99.6% at low (0.25–0.5), moderate (0.5–1) and high (1–2) citrate/Fe(II) molar ratios, respectively (Appendix A). During the oxidation process, the citrate complexed Fe(II) accounted for 18.7–44.5%, 37.0–77.2% and 63.6–93.1% of total Fe of oxygenation at low (0.25–0.5), moderate (0.5–1) and high (1–2) citrate/Fe(II) molar ratios (Figure 5g–i), respectively. Accordingly, the fractions of inorganic Fe^2+^ and adsorbed Fe(II) to total Fe of oxygenation decreased in the presence of citrate (Figure 5a–f). The increase of the fractions of Fe(II)-citrate complex with increasing citrate/Fe(II) molar ratio is in line with the observation that the high citrate/Fe(II) molar ratio was beneficial to •OH production (Figure 1 and Figure 2).

In summary, the influence of citrate/Fe(II) molar ratio on •OH production during Fe(II)-catalyzed O_2_ and H_2_O_2_ oxidation processes can be ascribed to the change of Fe(II)/Fe(III) speciation. In the absence of citrate, inorganic Fe(II)_dis_ and adsorbed Fe(II) contributed to •OH production. In the presence of citrate, the adsorbed Fe(II) became negligible because Fe(III) precipitates were hindered. Therefore, inorganic Fe(II)_dis_ and Fe(II)-citrate complex contributed collectively to •OH production. At a low citrate/Fe(II) molar ratio (<0.5), inorganic Fe(II)_dis_ mainly contributed to •OH production, followed by Fe(II)-citrate complex. At a high citrate/Fe(II) molar ratio (1–2), Fe(II)-citrate complex mainly contributed to •OH production. As the •OH yield from H_2_O_2_ decomposition by Fe(II)-citrate complex is much higher than that by inorganic Fe(II)_dis_ (Table 2), the net •OH production increased with increasing citrate/Fe(II) molar ratio during Fe(II)-catalyzed O_2_ and H_2_O_2_ oxidation processes.

### 3.5. Effect of Citrate/Fe(II) Molar Ratio on Phenol Degradation during Fe(II)-Catalyzed O_2_ Oxidation Process

The oxidative impact of •OH produced in a Fe(II)-citrate system toward environmental pollutants was evaluated using phenol as a model pollutant. In a Fe(II)-citrate (1:1) system, phenol concentration decreased 53.6% within 40 min (Figure 6a). When 100 mM 2-propanol (*k*_2-propanol, •OH_ = 2 × 10^9^ M^−1^ s^−1^ [27]) was added into the above system, phenol degradation was almost completely inhibited (Figure 6a). As 2-propanol was the scavenger for •OH, the inhibition confirmed that •OH was the main oxidant for phenol degradation. To verify the applicability of the kinetic model for predicting pollutant degradation, phenol instead of benzoate (reaction B1) was added. Results showed that the model-predicted time trajectories of phenol were in accordance with the experimental observations (Figure 6a), which confirmed the applicability of the kinetic model.

Based on the kinetic model, we further assessed the influence of the citrate/Fe(II) molar ratio on phenol degradation in a Fe(II)-citrate system at pH 7. Figure 6b shows that the phenol degradation efficiency rapidly increased from 1.1% to 80.1% when the citrate/Fe(II) molar ratio increased from 0 to 1.8, while it decreased gradually to 46.2% when the citrate/Fe(II) molar ratio reached 8. The dependence between phenol degradation efficiency and citrate/Fe(II) molar ratio may be explained by the fact that citrate can effectively facilitate •OH production from inorganic Fe(II)_dis_ oxidation (Figure 1) but also competed with phenol to consume •OH. A moderate molar ratio of citrate to Fe(II) was more favorable for phenol degradation during the Fe(II)-catalyzed O_2_ oxidation process.

## 4. Conclusions

This study investigated the influence of citrate/Fe(II) molar ratio on •OH production and the related environmental impacts during Fe(II)-catalyzed O_2_ and H_2_O_2_ oxidation processes. Results highlighted that the citrate/Fe(II) molar ratio controlled •OH production. In the absence of citrate, •OH is mainly produced from the oxidation of inorganic Fe(II)_dis_ and Fe(II)_ad_ by O_2_ and H_2_O_2_. Because the •OH yield from H_2_O_2_ decomposition by inorganic Fe(II)_dis_ and Fe(II)_ad_ was relatively low, •OH production was negligible. In the presence of citrate, the complexation of citrate resulted in the formation of Fe(II)-citrate complex, which can effectively decompose H_2_O_2_ to produce •OH. Hence, the addition of citrate significantly enhanced •OH production during Fe(II)-catalyzed O_2_ and H_2_O_2_ oxidation processes at pH 6–7.5. The variation of citrate/Fe(II) molar ratio changed Fe(II)/Fe(III) speciation and the fraction of Fe(II)-citrate complex, thereby affecting •OH production. With the increase of citrate/Fe(II) molar ratio, the fraction of Fe(II)-citrate complex increased, so •OH accumulation increased. However, for pollutant removal, a high concentration of citrate can also compete with the pollutant to consume •OH, thus weakening pollutant removal efficiency. Therefore, an appropriate ligand/Fe(II) molar ratio is crucial to achieve ligand-enhanced pollutant removal in the remediation of contaminated soil and groundwater.

## Figures and Tables

**Figure 1 ijerph-19-12977-f001:**
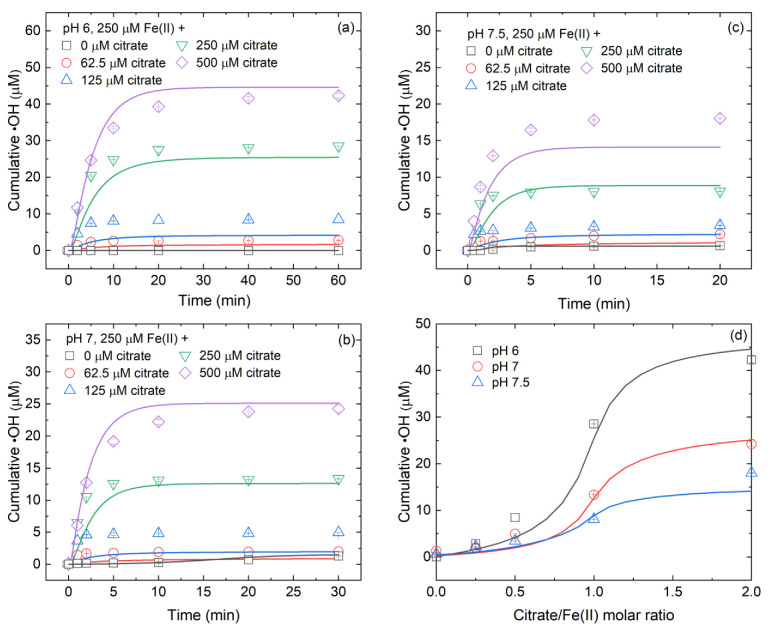
Effects of (**a**–**c**) citrate concentrations and (**d**) citrate/Fe(II) molar ratio on •OH production from oxygenation of inorganic Fe(II)_dis_. Initial conditions: variable citrate concentrations specified in panels (**a**–**c**), 250 μM Fe(II)_dis_, 20 mM benzoate and 10 mM buffer under oxic conditions. In panel (**d**), the calculations were based on 250 μM Fe(II)_dis_, 20 mM benzoate and 0.25 mM DO. The reaction times were set to be 60, 30, and 20 min for pH 6, 7 and 7.5, respectively. Points are the experimental results and lines are the modeled curves.

**Figure 2 ijerph-19-12977-f002:**
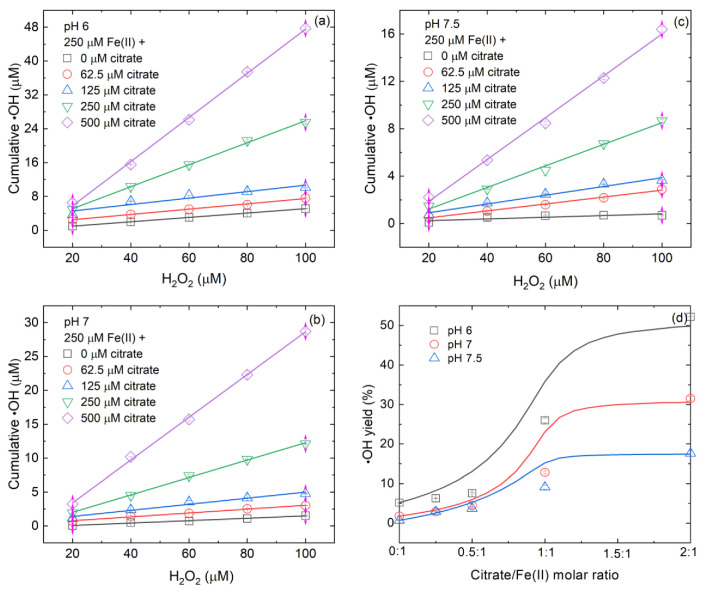
Effects of (**a**–**c**) citrate concentrations on •OH production from H_2_O_2_ decomposition by Fe(II)_dis_ and (**d**) the relationship between •OH yield relative to H_2_O_2_ decomposition and citrate/Fe(II) molar ratio. Initial conditions: variable citrate and H_2_O_2_ concentrations specified in panels (**a**–**c**), 250 μM Fe(II)_dis_, 20 mM benzoate and 10 mM buffer under anoxic conditions. Points are the experimental results. Lines are best fit linear regressions in panels (**a**–**c**) and are the modeled curves in panel (**d**) (for details, see Appendix A).

**Figure 3 ijerph-19-12977-f003:**
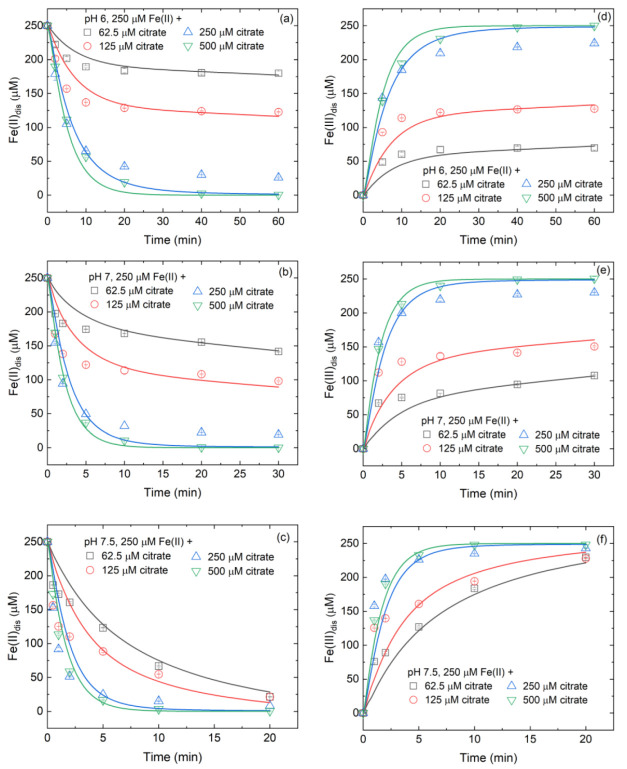
Effects of citrate concentrations on (**a**–**c**) Fe(II)_dis_ oxidation and (**d**–**f**) Fe(III)_dis_ generation in Fe(II)-citrate system. Initial conditions: variable citrate concentrations specified in panels (**a**–**f**), 250 μM Fe(II)_dis_, 20 mM benzoate and 10 mM buffer under oxic conditions. Points are the experimental results and lines are the modeled curves.

**Figure 4 ijerph-19-12977-f004:**
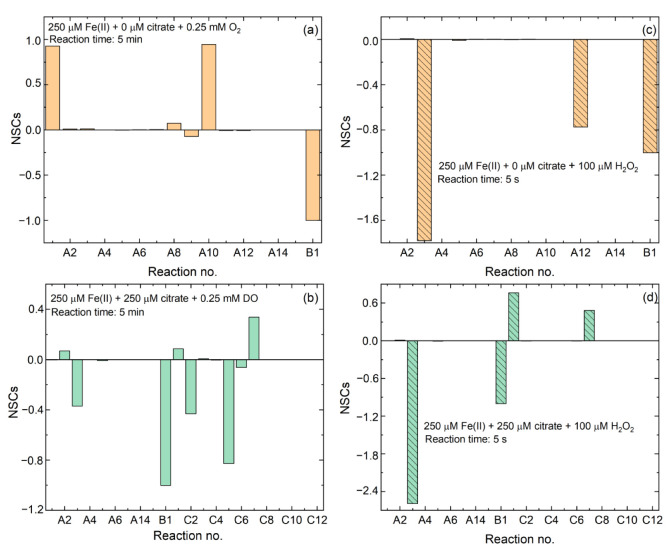
Normalized sensitivity coefficients (NSCs) for •OH production during Fe(II)-catalyzed (**a**,**b**) O_2_ and (**c**,**d**) H_2_O_2_ oxidation processes. Initial calculation conditions: Fe(II)_dis_, citrate and DO (or H_2_O_2_) concentrations specified, 20 mM benzoate and pH 7.

**Figure 5 ijerph-19-12977-f005:**
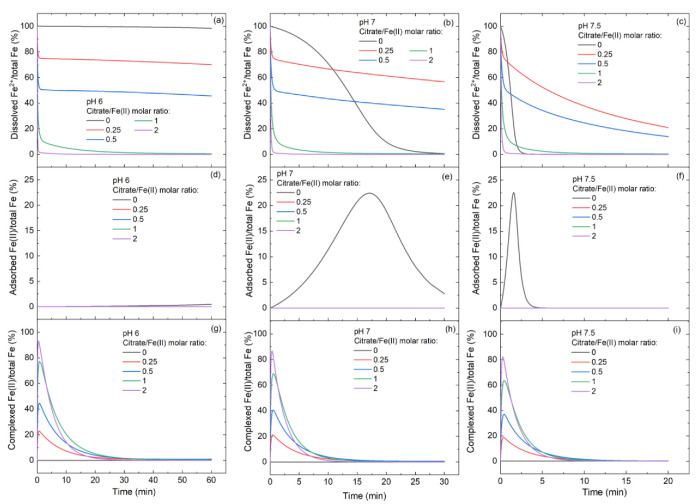
Modeled fractions of (**a**–**c**) dissolved Fe^2+^, (**d**–**f**) adsorbed Fe(II) and (**g**–**i**) Fe(II)-citrate complex during oxygenation of Fe(II)-citrate systems. Initial conditions: 250 μM Fe(II)_dis_ and 0.25 mM DO.

**Figure 6 ijerph-19-12977-f006:**
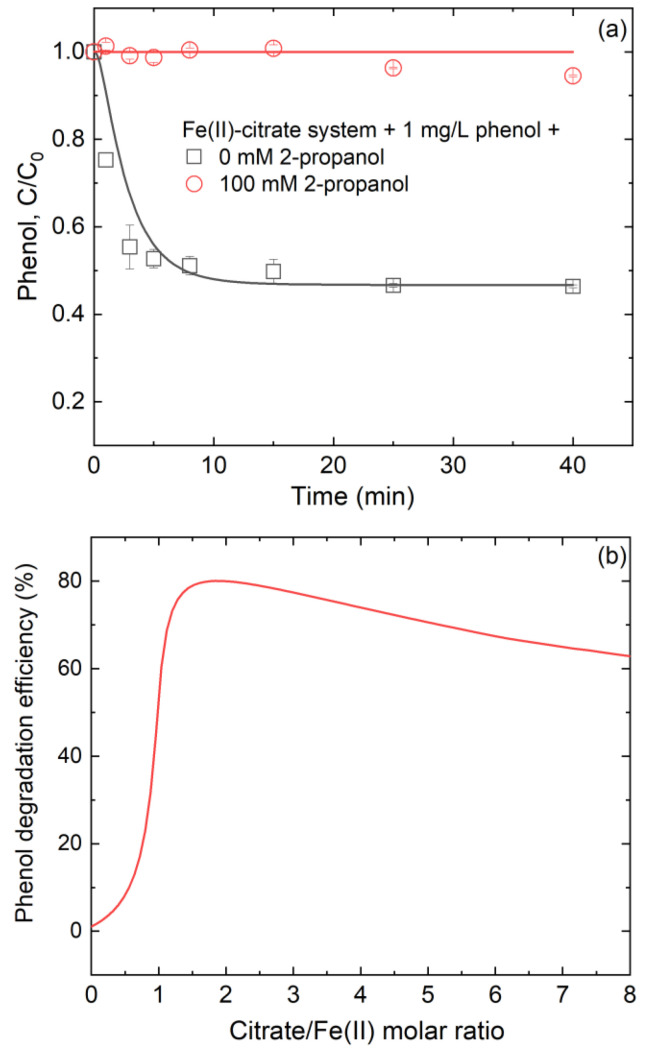
(**a**) Oxidation of phenol by the •OH produced in the Fe(II)-citrate system and (**b**) modeled variation of phenol degradation efficiency with citrate/Fe(II) molar ratio. In panel (**a**), the experimental conditions were based on 250 μM Fe(II)_dis_, 250 μM citrate and 1 mg/L phenol at pH 7; points are the experimental results and lines are the modeled curves. In panel (**b**), the calculations were based on 250 μM Fe(II)_dis_, 1 mg/L phenol, 0.25 mM DO and pH 7; citrate concentration was set based on the citrate/Fe(II) molar ratio.

**Table 1 ijerph-19-12977-t001:** Reaction network of kinetic model for inorganic Fe(II)_dis_ alone and Fe(II)_dis_-citrate system.

No.	Reactions	Rate Constant			Source
		pH 6	pH 7	pH 7.5	
**Reactions in Inorganic Fe(II)_dis_ System**					
A1	Fe(II) + O_2_ → Fe(III) + •O_2_^−^	1 × 10^−3^ M^−1^·s^−1^	1.3 × 10^−1^ M^−1^·s^−1^	1.8 M^−1^·s^−1^	Fitting
A2	Fe(II) + •O_2_^−^ → Fe(III) + H_2_O_2_	1 × 10^7^ M^−1^·s^−1^			[31]
A3	Fe(II) + H_2_O_2_ → Fe(III) + (0.052, 0.018, 0.007) •OH + OH^−^	5.5 × 10^1^ M^−1^·s^−1^	4.79 × 10^3^ M^−1^·s^−1^	1.33 × 10^4^ M^−1^·s^−1^	[32]
A4	Fe(II) + •OH → Fe(III) + OH^−^	5 × 10^8^ M^−1^·s^−1^			[32]
A5	Fe(III) + •O_2_^−^ → Fe(II) + O_2_	1.5 × 10^8^ M^−1^·s^−1^			[31]
A6 ^a^	Fe(III) + Fe(III) → LEP + LEP	3.2 × 10^5^ M^−1^·s^−1^	3.4 × 10^6^ M^−1^·s^−1^	5.0 × 10^6^ M^−1^·s^−1^	[33]
A7 ^a^	Fe(III) + LEP → LEP + LEP	3.2 × 10^5^ M^−1^·s^−1^	3.4 × 10^6^ M^−1^·s^−1^	5.0 × 10^6^ M^−1^·s^−1^	[33]
A8	Fe(II) + LEP → Fe(II)-LEP	1.1 × 10^6^ M^−1^·s^−1^	1.0 × 10^8^ M^−1^·s^−1^	1.0 × 10^8^ M^−1^·s^−1^	[34]
A9	Fe(II)-LEP → Fe(II) + LEP	2.3 × 10^3^ M^−1^·s^−1^			[34]
A10	Fe(II)-LEP + O_2_ → LEP + LEP_i_ + •O_2_^−^	2 M^−1^·s^−1^	6 M^−1^·s^−1^	60 M^−1^·s^−1^	Fitting
A11	Fe(II)-LEP + •O_2_^−^ → LEP + LEP_i_ + H_2_O_2_	1 × 10^7^ M^−1^·s^−1^			[31]
A12	Fe(II)-LEP + H_2_O_2_ → LEP + LEP_i_ + (0.052, 0.018, 0.007) •OH + OH^−^	5.5 × 10^1^ M^−1^·s^−1^	4.79 × 10^3^ M^−1^·s^−1^	1.33 × 10^4^ M^−1^·s^−1^	[32]
A13	Fe(II)-LEP + •OH → LEP + LEP_i_ + +OH^−^	5 × 10^8^ M^−1^·s^−1^			[32]
A14	LEP + H_2_O_2_ → LEP + H_2_O + 0.5O_2_	3.1 × 10^−2^ M^−1^·s^−1^			[35]
A15	LEP + •O_2_^−^ → Fe(II) + LEP + O_2_	6.5 × 10^−2^ M^−1^·s^−1^			[36]
Trapping of •OH by benzoate					
B1	benzoate + •OH → HBA + •O_2_^−^	5.9 × 10^9^ M^−1^·s^−1^			[27]
Extended reactions in Fe(II)-citrate systems					
C1	Fe(II) + citrate → Fe(II)-citrate^−^	5.0 × 10^2^ M^−1^·s^−1^			[37]
C2	Fe(II)-citrate^−^ → Fe(II) + citrate	2.0 × 10^−3^ s^−1^			[37]
C3	Fe(III) + citrate → Fe(III)-citrate	2.1 × 10^5^ M^−1^·s^−1^			[37]
C4	Fe(III)-citrate → Fe(III) + citrate	1.1 × 10^−4^ s^−1^			[37]
C5	Fe(II)-citrate^−^ + O_2_ → Fe(III)-citrate + •O_2_^−^	2.9 M^−1^·s^−1^	8 M^−1^·s^−1^	12 M^−1^·s^−1^	Fitting
C6	Fe(II)-citrate^−^ + •O_2_^−^ → Fe(III)-citrate + H_2_O_2_	1 × 10^7^ M^−1^·s^−1^			[31]
C7	Fe(II)-citrate^−^ + H_2_O_2_ → Fe(III)-citrate + (0.522, 0.315, 0.176) •OH + OH^−^	1.3 × 10^2^ M^−1^·s^−1^	8 × 10^2^ M^−1^·s^−1^	5 × 10^4^ M^−1^·s^−1^	Fitting
C8	Fe(II)-citrate + •OH → Fe(III)-citrate + OH^−^	5 × 10^8^ M^−1^·s^−1^			[32]
C9	Fe(III)-citrate + H_2_O_2_ → Fe(II)-citrate + •O_2_^−^ + 2H^+^	2.5 × 10^−3^ M^−1^·s^−1^			[24]
C10	Fe(III)-citrate + •O_2_^−^ → Fe(II)-citrate + O_2_	5.6 × 10^2^ M^−1^·s^−1^			[38]
C11	Fe(III)-citrate + •OH → Fe(III)-citrate_ox_ + •O_2_^−^	1.2 × 10^8^ M^−1^·s^−1^			[21]
C12	citrate + •OH → citrate_ox_ + •O_2_^−^	5.0 × 10^7^ M^−1^·s^−1^			[27]

^a^ LEP and LEP_i_ represent the reactive and nonreactive lepidocrocite, respectively. Because the presence of citrate inhibited the hydrolysis and precipitate of inorganic Fe(III)_dis_, the formation and oxidation of Fe(II)_ad_ (reactions A8–A13) were not included in the Fe(II)-citrate system and the rate constants of Fe(III) hydrolysis and precipitate (reactions A6–A7) were adjusted to <1 × 10^3^, <1 × 10^3^ and <1 × 10^4^ M^−1^ s^−1^ for pH 6, 7 and 7.5, respectively.

**Table 2 ijerph-19-12977-t002:** A summary of •OH yield from H_2_O_2_ decomposition by Fe(II)_dis_.

Experimental Conditions		Yield of •OH Relative to H_2_O_2_ Decomposition	R^2^
pH 6	250 μM Fe(II)_dis_	5.2 ± 0.03%	0.99
	250 μM Fe(II)_dis_ + 62.5 μM citrate	6.3 ± 0.02%	0.99
	250 μM Fe(II)_dis_ + 125 μM citrate	7.6 ± 0.2%	0.89
	250 μM Fe(II)_dis_ + 250 μM citrate	26.0 ± 0.6%	0.99
	250 μM Fe(II)_dis_ + 500 μM citrate	52.2 ± 0.01%	0.99
pH 7	250 μM Fe(II)_dis_	1.8 ± 0.04%	0.99
	250 μM Fe(II)_dis_ + 62.5 μM citrate	2.9 ± 0.01%	0.99
	250 μM Fe(II)_dis_ + 125 μM citrate	4.5 ± 0.4%	0.97
	250 μM Fe(II)_dis_ + 250 μM citrate	12.8 ± 0.3%	0.99
	250 μM Fe(II)_dis_ + 500 μM citrate	31.5 ± 0.5%	0.99
pH 7.5	250 μM Fe(II)_dis_	0.7 ± 0.02%	0.95
	250 μM Fe(II)_dis_ + 62.5 μM citrate	2.9 ± 0.01%	0.99
	250 μM Fe(II)_dis_ + 125 μM citrate	3.7 ± 0.3%	0.97
	250 μM Fe(II)_dis_ + 250 μM citrate	9.1 ± 0.5%	0.99
	250 μM Fe(II)_dis_ + 500 μM citrate	17.6 ± 0.7%	0.99

## Data Availability

The data will be available on request.

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
