# Peer review of "Quantification of the Influence of Citrate/Fe(II) Molar Ratio on Hydroxyl Radical Production and Pollutant Degradation during Fe(II)-Catalyzed O2 and H2O2 Oxidation Processes"

_ijerph, 2022, doi:10.3390/ijerph191912977_

Round 1
Reviewer 1 Report
The Authors provide a brief introduction to the subject matter and present the aim of the work which is associated with O2 and H2O2 activation to produce hydroxyl radicals in the organic pollutant degradation processes catalyzed by Fe(II) and ferric citrate involved. The Fe(II)/citrate molar ratio impact on active oxidant formation effectiveness in the pollutant (phenol) degradation processes was studied.
The submission is quite well prepared and I think it could be accept in present form after minor correction of the language (particularly with regard to the syntax and the wording of the text to enhance its clarity).
Author Response
Point 1: The Authors provide a brief introduction to the subject matter and present the aim of the work which is associated with O2 and H2O2 activation to produce hydroxyl radicals in the organic pollutant degradation processes catalyzed by Fe(II) and ferric citrate involved. The Fe(II)/citrate molar ratio impact on active oxidant formation effectiveness in the pollutant (phenol) degradation processes was studied.
The submission is quite well prepared and I think it could be accept in present form after minor correction of the language (particularly with regard to the syntax and the wording of the text to enhance its clarity).
Response 1: We thank the reviewer for the positive evaluation of our manuscript. As suggested, the English writing has been polished thoroughly in the manuscript.
Reviewer 2 Report
This manuscript reports on mathematical modeling and lab data fittings on the Fe(II)-O2/H2O2 processes in the presence and absence of model ligand citrate. Overall, it is well-designed and written manuscript. The follows are a few minor comments.
1. P1, L1 in the introduction, delete “on”; delete “which is” at the 4th line. P2, L5. “cheaper and easier”. Please limit such minor grammatical mistakes.
2. P1, L7 in the introduction. Please be more specific about such “specific conditions”.
3. P3, section 2.2, “To evaluate the environmental impact of •OH produced from oxygenation of Fe(II)-citrate system, …, dissolved O2 (DO) concentration was maintained at near 0.25 mM.” What does it mean? What is the purpose of this “environmental impact evaluation”? It seems the purpose of p-HBA is for hydroxyl radical quantification. If so, please indicate why it is chosen and how it is used in the introduction.
4. P4, section 2.5, L5. What is “Fe(II)ad”? It is explained in SI, but please also be specific in the main manuscript.
5. P6, last paragraph in section 3.1; P7, first paragraph in section 3.2. It is meaningless to list ALL numerical increases in Fig 1d. Only the first (or first two for section 3.2) and last sentences are useful. A paragraph like that in the end of section 3.2 is good.
6. P10, 2nd line at 2nd paragraph. Define “NSCs”.
7. P12, section 3.5, L3. “tablea”.
8. SI, Section S1, the last sentence. What does it mean by “roughly set” in “the •OH yield from H2O2 decomposition by citrate complexed Fe(II) can be roughly set to be 52.2%, 31.5% and 17.6% for pH 6, 7 and 7.5, respectively.”? Those numbers are shown in Table 1. Is there any correlation?
Author Response
This manuscript reports on mathematical modeling and lab data fittings on the Fe(II)-O2/H2O2 processes in the presence and absence of model ligand citrate. Overall, it is well-designed and written manuscript. The follows are a few minor comments.
Response: We appreciate the reviewer for the positive evaluation of our manuscript.
Specific comments
Point 1. P1, L1 in the introduction, delete “on”; delete “which is” at the 4th line. P2, L5. “cheaper and easier”. Please limit such minor grammatical mistakes.
Response: Done.
Point 2. P1, L7 in the introduction. Please be more specific about such “specific conditions”.
Response: Thanks for this concern. The specific conditions referred to Fe(II) exposure to O2 under circumneutral pH conditions, for example, when the reduced soil or sediment was exposed to oxic conditions. We have also revised the related statement.
Original sentence: “recent studies showed that oxygen (O2) can also act as •OH source in some specific conditions”.
Revised sentence: “recent studies showed that oxygen (O2) can also act as •OH source in some specific conditions, for example, when the reduced soil or sediment was exposed to oxic conditions”.
Point 3. P3, section 2.2, “To evaluate the environmental impact of •OH produced from oxygenation of Fe(II)-citrate system, …, dissolved O2 (DO) concentration was maintained at near 0.25 mM.” What does it mean? What is the purpose of this “environmental impact evaluation”? It seems the purpose of p-HBA is for hydroxyl radical quantification. If so, please indicate why it is chosen and how it is used in the introduction.
Response: Thanks for this concern. (1) The statements “To evaluate the environmental impact of •OH produced from oxygenation of Fe(II)-citrate system, 1 mg/L phenol (kphenol, •OH = 6.6×109 M-1s-1, [27]) was added to the Fe(II)-citrate solution containing 250 μM citrate, 250 μM Fe(II)dis and 100 mM boric acid. It is noted that boric acid had a marginal influence on pollutant degradation [5] due to the low reaction rate constant of boric acid with •OH.” were used to describe the pollutant degradation experiment, which is different from •OH production experiment. (2) The statement “The change of solution pH was less than 0.1 and the dissolved O2 (DO) concentration was maintained at near 0.25 mM” was used to describe the experimental conditions.
To clearly describe the experimental conditions, we revised the related statements and divided different experiments into different paragraphs. The statement “During the experimental process, the change of solution pH was less than 0.1 and the dissolved O2 (DO) concentration was maintained at near 0.25 mM” was revised to “During all the above experiments, the change of solution pH was less than 0.1 and the dissolved O2 (DO) concentration was maintained at near 0.25 mM”.
(3) The oxidation of benzoate to p-HBA was indeed used to quantify •OH production. As suggested, the statement “To quantify •OH production during the reaction process, the oxidation of benzoate to p-hydroxybenzoic acid (p-HBA) was used as a probe reaction [25, 26]” was added in the revised introduction.
Point 4. P4, section 2.5, L5. What is “Fe(II)ad”? It is explained in SI, but please also be specific in the main manuscript.
Response: “Fe(II)”ad represents the adsorbed Fe(II). To address the reviewer’s concern, we added the phrase “adsorbed Fe(II) (Fe(II)ad)” in the revised manuscript.
Point 5. P6, last paragraph in section 3.1; P7, first paragraph in section 3.2. It is meaningless to list ALL numerical increases in Fig 1d. Only the first (or first two for section 3.2) and last sentences are useful. A paragraph like that in the end of section 3.2 is good.
Response: As suggested, we removed the meaningless sentences in the revised manuscript.
Point 6. P10, 2nd line at 2nd paragraph. Define “NSCs”.
Response: The “NSCs” represent the normalized sensitivity coefficients. The positive NSC values mean that reactions produce •OH, while negative NSC values mean that reactions consume •OH.
To address the reviewer’s concern, we added the statement “The positive NSC values mean that reactions produce •OH, while negative NSC values mean that reactions consume •OH” in the revised manuscript.
Point 7. P12, section 3.5, L3. “tablea”.
Response: The phrase “tablea” was wrong and has been corrected to “Fig. 6a” in the revised manuscript.
Point 8. SI, Section S1, the last sentence. What does it mean by “roughly set” in “the •OH yield from H2O2 decomposition by citrate complexed Fe(II) can be roughly set to be 52.2%, 31.5% and 17.6% for pH 6, 7 and 7.5, respectively.”? Those numbers are shown in Table 1. Is there any correlation?
Response: Thanks for this concern. (1) Theoretically, pure citrate complexed Fe(II) should be used to determine the •OH yield from H2O2 decomposition by citrate complexed Fe(II). However, the pure citrate complexed Fe(II) was difficult to obtain because of the dissociation of citrate complexed Fe(II). A speciation calculation shows that the fraction of citrate complexed Fe(II) in total Fe(II) was near 99% when citrate/Fe(II) molar ratio was 2. For the sake of simplicity, we assumed that the •OH yield from H2O2 decomposition by citrate complexed Fe(II) was equal to the experimental measurements when citrate/Fe(II) molar ratio was 2. (2) The values of 52.2%, 31.5% and 17.6% were obtained from Table 1. To address the reviewer’s concern, the related discussion was added in the revised SI, Section S1.
Reviewer 3 Report
The manuscript titled “Quantification of the Influence of Citrate/Fe(II) Molar Ratio on Hydroxyl Radical Production and Pollutant Degradation during Fe(II)-Catalyzed O2 and H2O2 Oxidation Processes” was submitted by Bingbing Hu et al.
The manuscript is structure and written well. However, there are few short comings in this manuscript that must be improved in order to meet the standards of the journal. Specific comments are given below
Specific comments
1. Highlights your finding and significant results in abstract part.
2. Improve the conclusion part.
3. I didn’t find the any language errors.
4. All figures are good and high quality.
5. Results and discussion part is well explained.
Author Response
The manuscript titled “Quantification of the Influence of Citrate/Fe(II) Molar Ratio on Hydroxyl Radical Production and Pollutant Degradation during Fe(II)-Catalyzed O2 and H2O2 Oxidation Processes” was submitted by Bingbing Hu et al.
The manuscript is structure and written well. However, there are few short comings in this manuscript that must be improved in order to meet the standards of the journal. Specific comments are given below
Response: We appreciate the reviewer for the positive evaluation of our manuscript.
Specific comments
Point 1. Highlights your finding and significant results in abstract part.
Response: We appreciate this constructive comment. As suggested, we removed some unimportant statements while adding the discussion of the Fe(II) speciation at different citrate/Fe(II) molar ratios. After the modification, the abstract was more concise and focused.
Original abstract: Ligand-enhanced hydroxyl radical (•OH) production from Fe(II)-catalyzed O2 and H2O2 oxidation processes is an important strategy for enhancing the remediation of contaminated soils and groundwater. However, the influence of the molar ratio of ligands to Fe(II) on •OH production remains elusive. This study employed citrate and inorganic dissolved Fe(II) (Fe(II)dis) as the representative ligand and Fe(II) species, respectively, to quantitatively explore the influence of ligand/Fe(II) molar ratio on •OH production during Fe(II) oxidation by O2 and H2O2. Results showed that the presence of citrate significantly enhanced •OH production at pH 6–7.5 and the enhancement was highly dependent on the citrate/Fe(II) molar ratio. For the oxygenation of inorganic Fe(II), the •OH accumulation was <0.6–1.3 μM in the absence of citrate. However, the •OH accumulation increased to 2.0–8.5, 3.4–28.5 and 8.1–42.3 μM at low (0.25–0.5), moderate (0.5–1) and high (1–2) citrate/Fe(II) molar ratios, respectively. For the oxidation of Fe(II) by H2O2, the •OH yield relative to H2O2 decomposition was 0.7%–5.2% in the absence of citrate. However, the •OH yield rose to 2.9%–7.6%, 3.7%–26% and 9.1%–52.2% at low (0.25–0.5), moderate (0.5–1) and high (1–2) citrate/Fe(II) molar ratios, respectively. The enhancement mechanism was expectedly ascribed to the formation and oxidation of Fe(II)-citrate complex which can effectively decompose H2O2 to •OH. For pollutant degradation, 1 mg/L phenol was degraded by 53.6% within 40 min during oxygenation of Fe(II)-citrate system (1:1) at pH 7. According to the proposed mechanism, a kinetic model was developed to describe the redox reactions in Fe(II)-citrate systems and was used to quantify the influence of citrate/Fe(II) molar ratio on •OH production and pollutant degradation. Our results suggest that a moderate molar ratio of ligand/Fe(II) (0.5–1) may be optimal for Fe(II)-catalyzed O2 and H2O2 oxidation processes.
Revised abstract: Ligand-enhanced hydroxyl radical (•OH) production is an important strategy for Fe(II)-catalyzed O2 and H2O2 oxidation processes . However, the influence of the molar ratio of ligands to Fe(II) on •OH production remains elusive. This study employed citrate and inorganic dissolved Fe(II) (Fe(II)dis) as the representative ligand and Fe(II) species, respectively, to quantify this relationship. Results showed that •OH production was highly dependent on the citrate/Fe(II) molar ratio. For instance, for the oxygenation of Fe(II)dis, the •OH accumulations were 2.0–8.5, 3.4–28.5 and 8.1–42.3 μM at low (0.25–0.5), moderate (0.5–1) and high (1–2) citrate/Fe(II) molar ratios, respectively. At low citrate/Fe(II) molar ratio (<0.5), inorganic Fe(II)dis mainly contributed to •OH production, with the increase in the citrate/Fe(II) molar ratio to a high level (1–2), Fe(II)-citrate complex turned to the electron source for •OH production. The change in Fe(II) speciation with the increase of citrate/Fe(II) molar ratio elevated •OH production. For pollutant degradation, 1 mg/L phenol was degraded by 53.6% within 40 min during oxygenation of Fe(II)-citrate system (1:1) at pH 7. Our results suggest that a moderate molar ratio of ligand/Fe(II) (0.5–1) may be optimal for Fe(II)-catalyzed O2 and H2O2 oxidation processes.
Point 2. Improve the conclusion part.
Response: Thanks for this concern. As suggested, we added the statements “Results highlighted that the citrate/Fe(II) molar ratio controlled •OH production” and “With the increase of citrate/Fe(II) molar ratio, the fraction of Fe(II)-citrate complex increased, so •OH accumulation increased. However, for pollutant removal, the high concentration of citrate can also compete with pollutant to consume •OH, thus weakening pollutant removal efficiency” in the revised manuscript.
Point 3. I didn’t find the any language errors.
Response: We appreciate the reviewer for the positive evaluation of our manuscript.
Point 4. All figures are good and high quality.
Response: We appreciate the reviewer for the positive evaluation of our manuscript.
Point 5. Results and discussion part is well explained.
Response: We appreciate the reviewer for the positive evaluation of our manuscript.